# SARS-CoV-2 Spike Protein Enhances Carboxypeptidase Activity of Angiotensin-Converting Enzyme 2

**DOI:** 10.3390/ijms25116276

**Published:** 2024-06-06

**Authors:** Xóchitl Andrea Mendiola-Salazar, Melanie A. Munguía-Laguna, Martha Franco, Agustina Cano-Martínez, José Santamaría Sosa, Rocío Bautista-Pérez

**Affiliations:** 1Department of Molecular Biology, Instituto Nacional de Cardiología “Ignacio Chávez”, Mexico City 14080, Mexicomeli_aline31@hotmail.com (M.A.M.-L.); 2Carrera de Médico Cirujano, Facultad de Estudios Superiores Iztacala, Universidad Nacional Autónoma de México, Mexico City 54090, Mexico; 3Department of Cardio-Renal Pathophysiology, Instituto Nacional de Cardiología “Ignacio Chávez”, Mexico City 14080, Mexico; marthafranco@lycos.com (M.F.); mulosan@hotmail.com (J.S.S.); 4Department of Physiology, Instituto Nacional de Cardiología “Ignacio Chávez”, Mexico City 14080, Mexico; agustina.cano@cardiologia.org.mx

**Keywords:** ECA2, spike protein, diabetes, angiotensin II-induced hypertension

## Abstract

In this study, we investigated whether severe acute respiratory syndrome coronavirus-2 (SARS-CoV-2) spike protein may modify angiotensin-converting enzyme 2 (ACE2) activity in the plasma, heart, kidney, liver, lung, and six brain regions (amygdala, brain stem, cortex, hippocampus, hypothalamus, and striatum) of diabetic and hypertensive rats. We determine ACE2 activity in the plasma and lysates of heart, kidney, liver, lung, and six brain regions. MLN-4760 inhibits ACE2 activity in the plasma and all organs. On the other hand, soluble ACE2 (sACE2) activity increased in the plasma of diabetic rats, and there was no change in the plasma of hypertensive rats. ACE2 activity was augmented in the liver, brain stem, and striatum, while it decreased in the kidney, amygdala, cortex, and hippocampus of diabetic rats. ACE2 activity increased in the kidney, liver, and lung, while it decreased in the heart, amygdala, cortex, and hypothalamus of hypertensive rats. We measured the ACE2 content via enzyme-linked immunosorbent assay and found that ACE2 protein levels increased in the heart, while it decreased in the plasma, kidney, brain stem, cortex, hippocampus, hypothalamus, and striatum of diabetic rats. ACE2 protein levels decreased in the brain stem, cortex, hippocampus, and hypothalamus of hypertensive rats. Our data showed that the spike protein enhanced ACE2 activity in the liver and lungs of diabetic rats, as well as in the heart and three of the brain regions (cortex, hypothalamus, and striatum) of hypertensive rats.

## 1. Introduction

Coronavirus disease 2019 (COVID-19) is caused by severe acute respiratory syndrome coronavirus-2 (SARS-CoV-2). It is not yet clear what factors determine COVID-19 severity. In this regard, comorbidities such as cardiorenal disease, chronic obstructive pulmonary disease, obesity, diabetes, and hypertension correlate with COVID-19 severity [1]. Angiotensin-converting enzyme 2 (ACE2) is the receptor for SARS-CoV-2 and other coronaviruses (SARS-CoV, HCoV-NL63) [2,3]. Therefore, COVID-19 severity may be related to ACE2 through the following mechanisms. (1) ACE2 is expressed in the lungs and other organs (brain, heart, kidney, liver, lung, intestine, and vasculature). Thus, SARS-CoV-2 may invade them all [4,5,6,7]. (2) ACE2 overexpression in the organs may make them more vulnerable to SARS-CoV-2 [8]. (3) ACE2 is involved in the cleavage of vasoactive peptides and the activation of several systems (coagulation, kinin–kallikrein, and the renin–angiotensin system) causing damage to the lungs; endothelial, neurological, kidney, and cardiovascular failure; coagulopathy; and inflammation [9]. (4) The liberation of soluble ACE2 (sACE2) may act as a receptor for SARS-CoV-2 and inhibit the binding of SARS-CoV-2 to the ACE2 of membranes [10]. 

As mentioned earlier, ACE2 is an enzyme that not only cleaves angiotensin II (Ang II) to Ang (1–7) but also removes the carboxyl-terminal residue from another peptide: apelin-13, apelin-36, [des-Arg]^9^ bradykinin ([des-Arg]^9^ BK), Lys [des-Arg]^9^ BK, neurotensin (1–8), Dynorphin A (1–13), and β-Casomorphin-(1–7) [11,12]. In addition, numerous studies have shown that metalloproteinase 17 (ADAM17) cleaves the catalytically active ectodomain of ACE2 and releases sACE2 in serum, plasma, and urine [13,14]. In this regard, sACE2 activity increases in the plasma of patients with heart failure, myocardial infarction, and hypertension [15,16,17,18]. In plasma and tissues, Ang (1–7) results from the degradation of Ang II by ACE2 [19]. Interestingly, the serum of diabetic and obese patients shows increased [des-Arg]^9^ BK levels, while the serum of hypertensive patients shows increased Ang (1–7) levels [20,21]. This evidence suggests that ACE2 down-regulation impairs the inactivation of [des-Arg]^9^ BK, resulting in an inflammatory cascade, which leads to increased cytokine release in COVID-19 [22]. ACE2 knockout mice and ACE2 pharmacological inhibition potentiate the vasodepressor effect of pyr-apelin 13 and apelin 17 [23]. 

As mentioned earlier, ACE2 is the receptor of SARS-CoV-2. The spike protein of SARS-CoV-2 contains the receptor-binding domain (RBD) in the S1 subunit [24,25]. Recently, in vitro experiments have shown that the spike protein increases recombinant human ACE2 (rhACE2) activity against [des-Arg]^9^ BK. In contrast, the cleavage of Ang II was slightly affected by the interaction between the spike protein and ACE2 [26,27]. These findings suggest that the spike protein of SARS-CoV-2 may affect ACE2 activity in the plasma and organs of diabetic or hypertensive patients, thus contributing to the severity of COVID-19.

Therefore, in this study, we used two experimental models, one for diabetes and the other for hypertension. To evaluate the participation of ACE2 (SARS-CoV-2 receptor) in these experimental models, we determined the carboxypeptidase activity and the concentration of ACE2 in organs that may be sensitive to SARS-CoV-2. Also, we investigated whether the spike protein of SARS-CoV-2 modifies ACE2 activity in the plasma and organ lysate of diabetic and hypertensive rats. Thus, we relate several factors that contribute to the severity of COVID-19.

## 2. Results

### 2.1. Blood Glucose and Systolic Blood Pressure (SBP)

The blood glucose concentration in diabetic rats was 578 ± 24 mg/dL, and in control rats, it was 116 ± 2.5 mg/dL. The SBP in hypertensive rats was 196 ± 10 mm Hg, and in control rats, it was 123 ± 5 mm Hg.

### 2.2. Activity of Angiotensin-Converting Enzyme 2 (ACE2)

First, we evaluate specific ACE2 activity in the lysates of heart, kidney, liver, lung, six brain regions, and plasma with and without MNL4760, an ACE2 inhibitor. Figure 1 and Figure 2 show that MNL4760 inhibits ACE2 activity in the lysates of the heart, kidney, liver, lung, and plasma of the control, diabetic, and hypertensive groups. Figure 3 and Figure 4 show that MNL4760 inhibits ACE2 activity in amygdala, brain stem, cortex, hippocampus, hypothalamus, and striatum of the control, diabetic, and hypertensive rats.

ACE2 basal activity was as follows, from highest to lowest: lung, kidney, liver, heart, and plasma. Figure 5 also shows ACE2 activity with and without SARS-CoV-2 spike protein in the plasma and organs of diabetic rats. Under these conditions, we observed that in the plasma of diabetic rats, ACE2 activity increased compared to the control group, and there were no changes in the presence of the spike protein (Figure 5A). In the heart, ACE2 activity did not change in the absence or presence of the spike protein in both control and diabetic groups (Figure 5B). In the kidney, ACE2 activity decreased in diabetic rats compared to the control group, and there were no changes in the presence of the spike protein (Figure 5C). In the liver, ACE2 activity increased in diabetic rats compared to the control group and increased even more in the presence of the spike protein (Figure 5D). In the lung, ACE2 activity in diabetic rats did not change compared to the control group but increased in the presence of the spike protein (Figure 5E).

ACE2 basal activity in six brain regions was as follows, from highest to lowest: amygdala, hippocampus, cortex, hypothalamus, striatum, and brain stem. Figure 6 shows ACE2 activity with and without SARS-CoV-2 spike protein in six brain regions of diabetic rats. Under these conditions, we observed that in the hypothalamus, ACE2 activity did not change in the absence or presence of the spike protein compared to the control group (Figure 6A). In the amygdala, hippocampus, and cortex, ACE2 activity decreased compared to the control group, and there were no changes in the presence of the spike protein (Figure 6B,C,E). ACE2 activity in the striatum and brain stem increased compared to the control group and did not change with and without the spike proteins of SARS-CoV-2 (Figure 6D,F).

Figure 7 shows ACE2 activity in the absence and presence of SARS-CoV-2 spike proteins in the plasma and organs of hypertensive rats. Under these conditions, in the plasma, ACE2 activity did not change with and without spike proteins compared to the control group (Figure 7A). In the heart, ACE2 activity decreased compared to the control group but increased in the presence of the spike protein (Figure 7B). In the kidney, liver, and lung, ACE2 activity increased compared to the control group, and there were no changes in the presence of the spike protein (Figure 7C–E).

Figure 8 shows ACE2 activity in six brain regions of hypertensive rats with and without SARS-CoV-2 spike protein. In the hypothalamus, ACE2 activity decreased compared to the control group but increased in the presence of the spike protein (Figure 8A). In the amygdala, ACE2 activity decreased compared to the control group, and there were no changes in the presence of the spike protein (Figure 8B). In the hippocampus, ACE2 activity did not change in the absence or presence of the spike protein compared to the control group (Figure 8C). In the striatum, ACE2 activity did not change compared to the control group but increased in the presence of the spike protein (Figure 8D). In the cortex, ACE2 activity decreased compared to the control group but increased in the presence of the spike protein (Figure 8E). In the brain stem, ACE2 activity did not change in the absence or presence of the spike protein compared to the control group (Figure 8F).

Figure 9 shows that the content of ACE2 protein decreased in the plasma and kidney, increased in the heart, and was not modified in the liver and lungs of diabetic rats compared to the control group. Figure 10 shows that the content of ACE2 protein decreased in the hypothalamus, hippocampus, striatum, cortex, and brain stem and was not modified in the amygdala of the brain of diabetic rats compared to the control group. Figure 11 shows that the content of ACE2 protein was not altered in the plasma, heart, kidney, liver, and lungs of hypertensive rats compared to the control group. Figure 12 shows that the content of ACE2 protein decreased in the hypothalamus, hippocampus, cortex, and brain stem and was not modified in the amygdala and striatum of the brain of hypertensive rats compared to the control group.

## 3. Discussion

The severity of COVID-19 is related to diseases such as obesity, diabetes, and hypertension, as well as ACE2 expression in diverse organs, and it has been recently suggested that the spike proteins of SARS-CoV-2 may also affect ACE2 activity [26,27].

It is well-known that hypertension and diabetes can affect diverse organs such as the brain, heart, lungs, kidneys, vessels, and liver [28,29,30]. Numerous studies have demonstrated that ACE2 protects these organs in pathophysiological conditions [31,32,33,34,35]. However, ACE2 expression in these organs may make them more vulnerable to the interaction between SARS-CoV-2 and ACE2.

In this study, we evaluated whether the spike protein of SARS-CoV-2 modifies ACE2 activity in the plasma, heart, kidney, liver, lung, and brain regions (amygdala, brain stem, cortex, hippocampus, hypothalamus, and striatum) of diabetic and hypertensive rats. These organs participate in the pathophysiology of metabolic and cardiovascular diseases, such as diabetes and hypertension. Moreover, in the brain, the regions of the limbic system (amygdala, brain stem, cortex, hippocampus, hypothalamus, and striatum) are affected by diabetes and hypertension. In this regard, several studies have shown alterations in the cerebral vasculature that cause nerve damage and lead to cognitive impairment in diabetes and hypertension [36,37,38].

It is also important to mention that ACE2 and angiotensin-converting enzyme (ACE) have ~40% homology in the amino acid sequence. In addition, ACE and ACE2 have different inhibitors and substrates. ACE inhibitors do not inhibit the monocarboxypeptidase activity of ACE2 [12]. A recent study evaluated the activity of ACE2 with a fluorogenic substrate synthesized by them (like Ang II, apelin 13, [des-Arg]^9^ BK, dynorphin A, and Ang I) and also used a commercial fluorogenic pseudosubstrate [27].

We evaluated the carboxypeptidase activity of ACE2, for which we used Mca-Ala-Pro-Lys (Dnp) (commercial fluorogenic substrate). Our data show that ACE2 basal activity was as follows, from highest to lowest: lung, kidney, liver, heart, and plasma. ACE2 activity in brain regions was as follows, from highest to lowest: amygdala, hippocampus, cortex, hypothalamus, striatum, and brain stem.

To demonstrate that the activity corresponds to ACE2 and not ACE, we used MNL-4760, a specific ACE2 inhibitor. Our data indicate that the MLN-4760 inhibits ACE2 activity in the plasma and all organs. A recent study showed that the inhibitor MNL-4760 binds to the active site of ACE2 [39].

In addition, our results demonstrate that ACE2 activity does not depend on the amount of protein. In the lungs, there are no changes in the activity and protein levels of ACE2. The activity and protein levels of ACE2 decrease in the kidney, brain cortex, and hippocampus. In the heart, the activity does not change, but the ACE2 protein level increases. In the liver, the activity but not the protein level of ACE2 increases. In the amygdala, the activity decreases, and the ACE2 protein level does not change. In the hypothalamus, the activity does not change, but the ACE2 protein level increases. In the striatum and brain stem, the activity increases, but the ACE2 protein level decreases. We observed these results in diabetic rats.

In the kidney, liver, and lungs, the activity increased but not the ACE2 protein level. In the heart and amygdala, the activity decreased, but there was no change in the ACE2 protein. In the brain stem and hippocampus, the ACE2 activity did not change but the ACE2 protein level decreased.

In the cortex and hypothalamus, the activity and protein level of ACE2 decreased. In the striatum, there were no changes in the activity and protein level of ACE2. We observed these results in hypertensive rats.

sACE2 activity (plasma) increases but sACE2 protein level decreases in diabetic rats, and there was no change in hypertensive rats. These results suggest that ACE2 activity did not correlate with ACE2 protein level in some organs nor with sACE2 activity or sACE2 protein level (plasma) in diabetic and hypertensive rats. This observation is in agreement with other studies [40,41,42,43,44,45,46,47,48,49].

Our results show that ACE2 protein level decreased in the brain stem, cortex, hippocampus, and hypothalamus of diabetic and hypertensive rats. Also, ACE2 protein decreased in the striatum of diabetic rats. In this regard, diabetes and hypertension increase the permeability of the blood–brain barrier, which facilitates the entry of Ang II to the brain stem and hypothalamus, which are brain regions that regulate blood pressure [50,51]. In addition, the activation of the AT1 receptor or kinin B1 receptor promotes the shedding of ACE2 in the cellular membrane and reduces ACE2 activity in hypothalamus neurons [52,53]. In the hypothalamus of DOCA-salt hypertensive mice, ACE2 activity and expression decrease, while ACE2 activity increases in the cerebrospinal fluid, possibly due to ADAM17 releasing sACE2, which is catalytically active [54]. The post-translational modifications that regulate the ACE2 protein are the shedding of the ACE2 ectodomain, ubiquitination, phosphorylation, and glycosylation.

In this regard, Ang II downregulates ACE2 expression in a human kidney tubular epithelial cell line (HK-2), neurons, and astrocytes [55,56,57]. Ang II also regulates the internalization and degradation of ACE2 protein in lysosomes. The internalization involves ubiquitination through an AT1 receptor-dependent mechanism [58,59]. ACE2 phosphorylation enhances the stability of ACE2 [60]. Interestingly, the N-glycosylation of ACE2 regulates the interaction between the spike protein of SARS-CoV-2 and ACE2 [61,62,63].

A recent study reports that in experiments in vitro, the spike protein of SARS-CoV-2 increased recombinant human ACE2 activity against des-Arg^9^-BK [26,27].

We determine ACE2 activity in the plasma and lysates of organs in the presence of the spike protein of SARS-CoV-2. Our data showed that ACE2 activity increases the presence of the spike protein in the liver and lungs of diabetic rats and in the heart and three regions of the brain (cortex, hypothalamus, and striatum) of hypertensive rats.

Rat models have been used to study the pathophysiology of human diseases. In this regard, the ACE2 protein of rats interacts weakly with spike protein of SARS-CoV, possibly because rat ACE2 contains glycosylation sites that lead to steric interference with SARS-CoV binding [64,65,66]. Understanding this mechanism can help in the search for therapeutic alternatives.

Also, it is necessary to consider the participation of ACE2 in the metabolism of other peptides of physiological importance, and not only to Ang II and [des-Arg]^9^ BK. Apelin is a cardioprotective peptide, and its loss contributes to systolic dysfunction and heart failure [23,67,68]. Dynorphin A (1–13) is also a protective peptide in pulmonary hypertension [69]. Neurotensin is a neuromodulator in the amygdala and other regions of the central nervous system [70]. Then, the spike protein can enhance ACE2 activity and increases the hydrolysis of protective peptides such as apelin, dynorphin A (1–13), and neurotensin. Possibly, this mechanism contributes to the complications that occur in COVID-19.

## 4. Materials and Methods

### 4.1. Animal Models

The procedures used in this study were performed in accordance with the Mexican Federal Regulation for Animal Experimentation and Care (NOM-062- ZOO-1999, published in 2001). This study was approved by the Institutional Committee for the Care and Use of Laboratory Animals of the Instituto Nacional de Cardiología “Ignacio Chávez” under protocol number INC-CICUAL/002/2023.

#### 4.1.1. Streptozotocin-Induced Diabetic Rats

Male Wistar rats (300–350 g) were divided into two groups (n = 10 each): (1) control and (2) streptozotocin (STZ)-induced diabetic rats. The animals with blood glucose of 300 mg/dl (18 mmol/L) were used as diabetic rats and maintained for 30 days, as we previously reported [71].

#### 4.1.2. Angiotensin II-Induced Hypertension Rats

Male Wistar rats (350–360 g) were divided into two groups (n = 10 each): (1) control and (2) hypertensive rats (435 ng/kg/min of Ang II, which was infused through osmotic minipumps for 14 days (Alzet 2002; Alza, Palo Alto, CA, USA), as we previously reported [71].

#### 4.1.3. Sample Collection and Tissue Preparation

The rats were anesthetized with pentobarbital sodium (30 mg/kg, i.p), the blood was collected in heparin tubes, and the plasma was obtained via centrifugation and stored at −80 °C until use. Before dissection, organs (heart, kidney, liver, lung, and brain) were perfused with ice-cold phosphate-buffered solution (PBS), pH 7.4, frozen in liquid nitrogen, and stored at −80 °C until use.

### 4.2. Determination of ACE2 Activity

Organs (100 mg) were homogenized with 500 μL of buffer (75 mM Tris-HCl, pH 7.5, 1 M NaCl, 0.5 mM ZnCl_2_, 0.5% Triton X-100, and protease inhibitors (EDTA-free)) on ice. The homogenate was centrifuged at 12,000× *g* for 15 min at 4 °C. The supernatant was aliquoted and stored at −80 °C until use. The protein concentration was determined using Bradford assays [72,73].

ACE2 activity was carried out in 96-well plates; the assay was carried in a 100 μL total volume at 37 °C: 45 µL of buffer (75 mM Tris-HCl, pH 7.5, 1 M NaCl, 0.5 mM ZnCl_2_), 5 µL of sample (10 µg of protein), and 50 µL of 10 µM Mca-APK (DnP) substrate solution (a specific ACE2 quenched fluorogenic substrate) (Enzo Life Sciences, Inc., Farmingdale, NY, USA).

To determine the specific activity of ACE2, the samples were incubated with and without 10 mM MLN-4760 (an ACE2 inhibitor) for 30 min at room temperature before being added to the ACE2 substrate.

To test the effect of the spike protein on ACE2 activity, we measured ACE2 activity with and without 10 µg of spike protein (SARS-CoV-2 Spike Glycoprotein Receptor Binding Motif Item No. 30428, Cayman, UK). To determine this amount of spike protein, ACE2 activity was carried out in lung samples in the presence of different amounts of spike protein. The graph is shown in the Appendix A.

ACE2 activity was expressed in relative fluorescent units (RFUs) at λex 320 nm/λem 420 nm. Readings were recorded every 5 min for 30 min to 3 hours at room temperature (Synergy^®^HTX multimode BioTek^®^ Instruments, Inc., Winooski, VT, USA). To calculate ΔRFU/ΔT, two time points were chosen (T1 and T2) in the linear range of the plot, and the corresponding values for fluorescence were obtained (RFU1 and RFU2).

### 4.3. Enzyme-Linked Immunosorbent Assay (ELISA) for ACE2 Determination

ACE2 level was measured by a commercial sandwich enzyme-linked immunosorbent assay (ELISA) kit following the manufacturer’s instructions (ER0609, Fine Test^®^, Wuhan, Hubei, China) [74]. We used 30 μg of protein for each of the samples and added it to a pre-coated antibody. These assays were read at 450 nm (Synergy^®^ HTX multimode BioTek^®^ Instruments, Inc., Winooski, VT, USA). The concentration of ACE2 in the tested samples was estimated against the standard curve. Concentrations are reported as ng/mg of protein.

### 4.4. Statistical Analysis

Values were expressed as the x̄ ± SE. Statistical differences among groups were calculated using a one-way ANOVA followed by the Bonferroni test using GraphPad Prism 8 software (GraphPad, San Diego, CA, USA); *p* < 0.05 was considered statistically significant.

## 5. Conclusions

Our results suggest that the SARS-CoV-2 spike protein enhances the carboxypeptidase activity of ACE2 in some organs of diabetic and hypertensive rats. It is possible that rat ACE2 presents changes in its structure that favor or prevent its interaction with the SARS-CoV-2 spike protein.

## Figures and Tables

**Figure 1 ijms-25-06276-f001:**
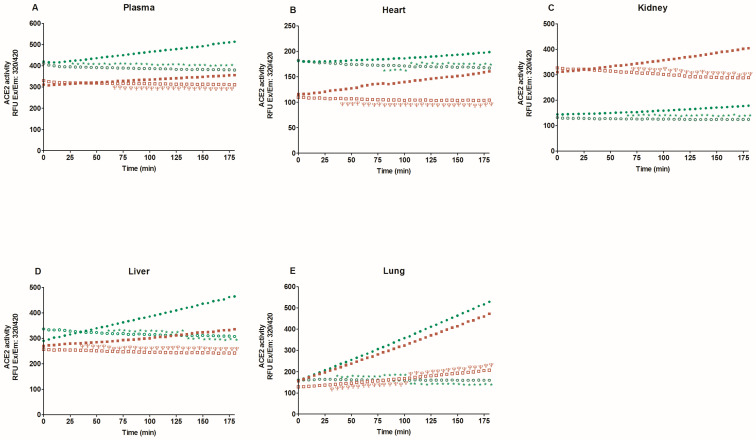
ACE2 activity in the plasma, heart, kidney, liver, and lung of control and diabetic rats. ACE2 activity was determined for three hours with and without MNL4760. The fluorescent intensities (RFU) were plotted as a function of the reaction time. The data are $\bar{x}$ ± SE of *n* = 10 with 3 replicates each. ^Ψ^
*p* < 0.05 control + MNL4760 (□) vs. control (■) and * *p* < 0.05 diabetic + MNL4760 (○) vs. diabetic (●).

**Figure 2 ijms-25-06276-f002:**
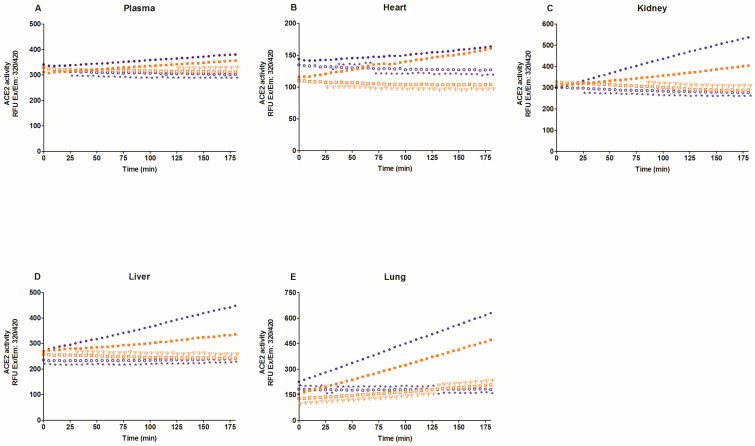
ACE2 activity in the plasma, heart, kidney, liver, and lung of control and hypertensive rats. ACE2 activity was determined for three hours with and without MNL4760. The fluorescent intensities (RFU) were plotted as a function of the reaction time. The data are $\bar{x}$ ± SE of *n* = 10 with 3 replicates each. ^Ψ^
*p* < 0.05 control + MNL4760 (□) vs. control (■) and * *p* < 0.05 hypertensive + MNL4760 (○) vs. hypertensive (●).

**Figure 3 ijms-25-06276-f003:**
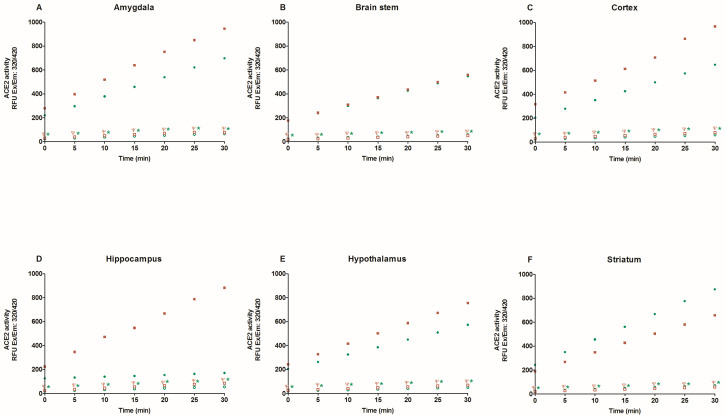
ACE2 activity in amygdala, brain stem, cortex, hippocampus, hypothalamus, and striatum of control and diabetic rats. ACE2 activity was determined for thirty minutes with and without MNL4760. The fluorescent intensities (RFU) were plotted as the function of the reaction time. The data are $\bar{x}$ ± SE of *n* = 10 with 3 replicates each. ^Ψ^
*p* < 0.05 control + MNL4760 (□) vs. control (■) and * *p* < 0.05 diabetic + MNL4760 (○) vs. diabetic (●).

**Figure 4 ijms-25-06276-f004:**
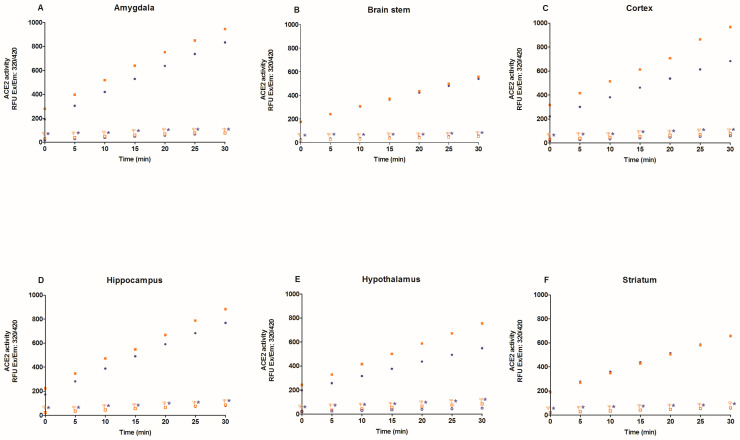
ACE2 activity in amygdala, brain stem, cortex, hippocampus, hypothalamus, and striatum of control and hypertensive rats. ACE2 activity was determined for thirty minutes with and without MNL4760. The fluorescent intensities (RFU) were plotted as the function of the reaction time. The data are $\bar{x}$ ± SE of *n* = 10 with 3 replicates each. ^Ψ^
*p* < 0.05 control + MNL4760 (□) vs. control (■) and * *p* < 0.05 hypertensive + MNL4760 (○) vs. hypertensive (●).

**Figure 5 ijms-25-06276-f005:**
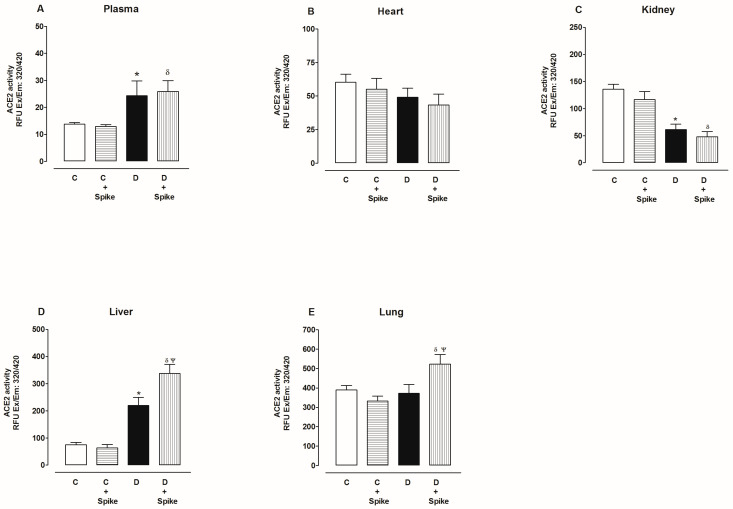
ACE2 activity in the plasma, heart, kidney, liver, and lung of control and diabetic rats. ACE2 activity was determined in the absence and presence of spike proteins of SARS-CoV-2. The data are $\bar{x}$ ± SE of *n* = 10. * *p* < 0.05 Diabetic (D) vs. Control (C), ^δ^
*p* < 0.05 D + spike protein vs. C + spike protein, ^Ψ^
*p* < 0.05 D + spike protein vs. D.

**Figure 6 ijms-25-06276-f006:**
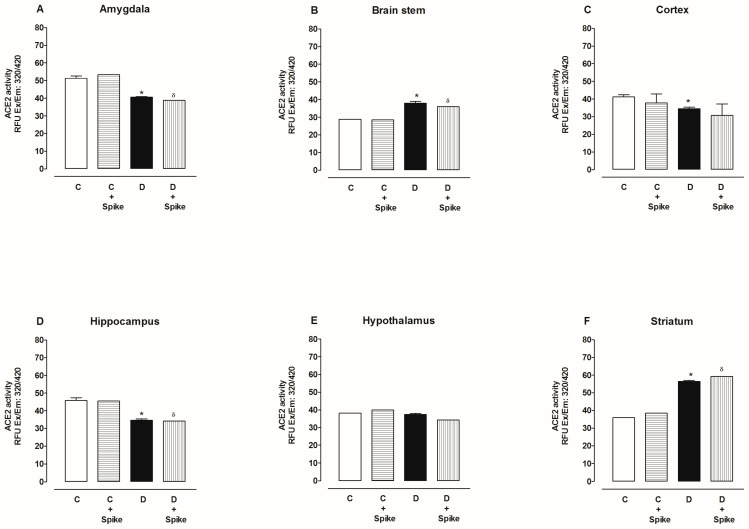
ACE2 activity in amygdala, brain stem, cortex, hippocampus, hypothalamus, and striatum of control and diabetic rats. ACE2 activity was determined in the absence and presence of spike proteins of SARS-CoV-2. The data are $\bar{x}$ ± SE of *n* = 10. * *p* < 0.05 Diabetic (D) vs. (C), ^δ^
*p* < 0.05 D + spike protein vs. C + spike protein.

**Figure 7 ijms-25-06276-f007:**
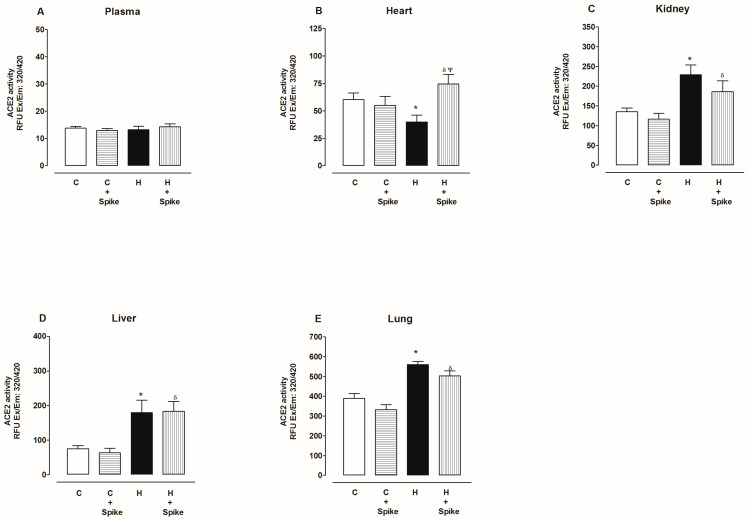
ACE2 activity in the plasma, heart, kidney, liver, and lung of control and hypertensive rats. ACE2 activity was determined in the absence and presence of spike proteins of SARS-CoV-2. The data are $\bar{x}$ ± SE of *n* = 10. * *p* < 0.05 Hypertensive (H) vs. control (C), ^δ^
*p* < 0.05 H + spike protein vs. C + spike protein, ^Ψ^
*p* < 0.05 H + spike protein vs. H.

**Figure 8 ijms-25-06276-f008:**
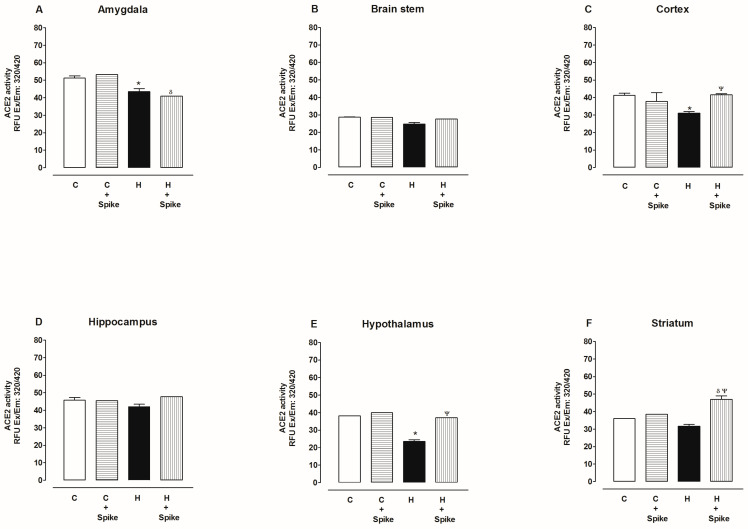
ACE2 activity in the amygdala, brain stem, cortex, hippocampus, hypothalamus, and striatum of control and hypertensive rats. ACE2 activity was determined in the absence and presence of spike proteins of SARS-CoV-2. The data are $\bar{x}$ ± SE of *n* = 10. * *p* < 0.05 Hypertensive (H) vs. control (C), ^δ^
*p* < 0.05 H + spike protein vs. C + spike protein, ^Ψ^
*p* < 0.05 H + spike protein vs. H.

**Figure 9 ijms-25-06276-f009:**
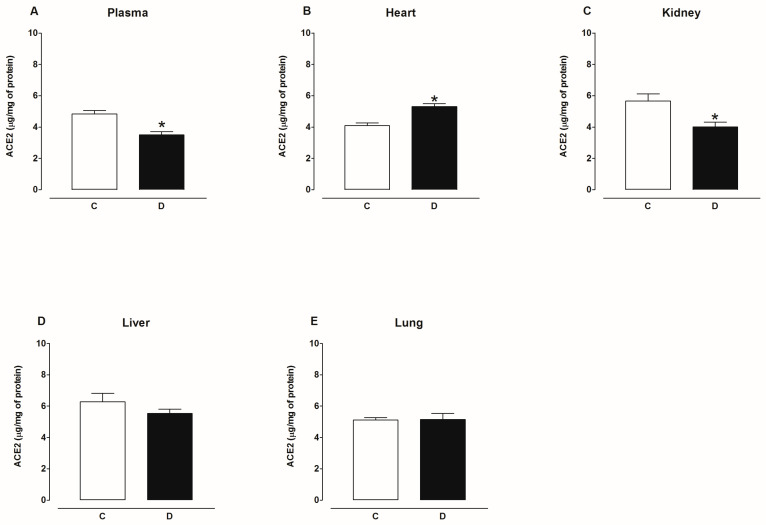
ACE2 content in plasma, heart, kidney, liver, and lung of control (C) and diabetic (D) rats. Each bar represents the $\bar{x}$ ± SE of *n* = 10. * *p* < 0.05 when compared with control.

**Figure 10 ijms-25-06276-f010:**
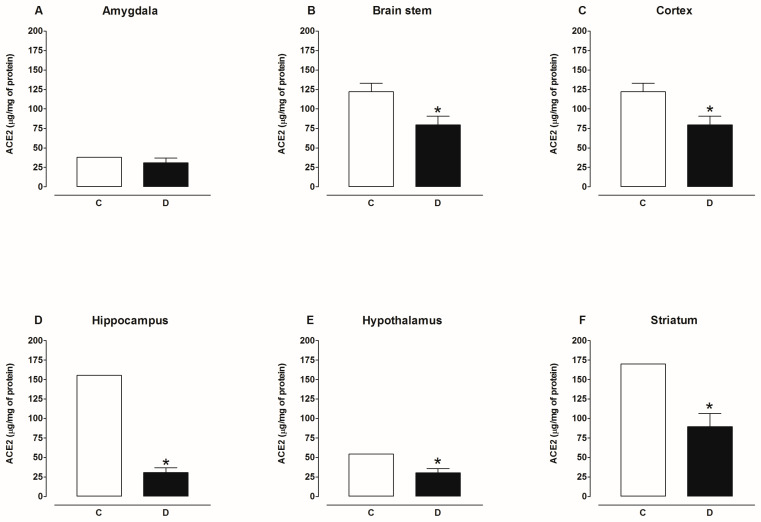
ACE2 content in amygdala, brain stem, cortex, hippocampus, hypothalamus, and striatum of control (C) and diabetic (D) rats. Each bar represents the $\bar{x}$ ± SE of *n* = 10. * *p* < 0.05 when compared with control.

**Figure 11 ijms-25-06276-f011:**
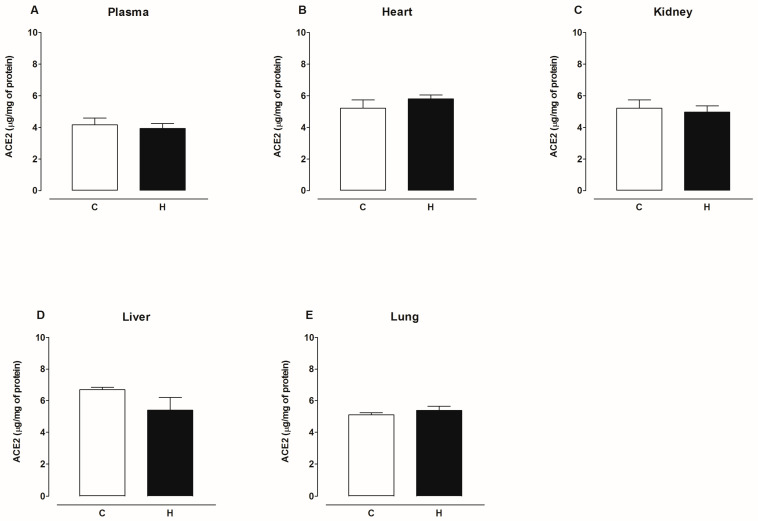
ACE2 content in plasma, heart, kidney, liver, and lung of control (C) and hypertensive (H). Each bar represents the $\bar{x}$ ± SE of *n* = 10.

**Figure 12 ijms-25-06276-f012:**
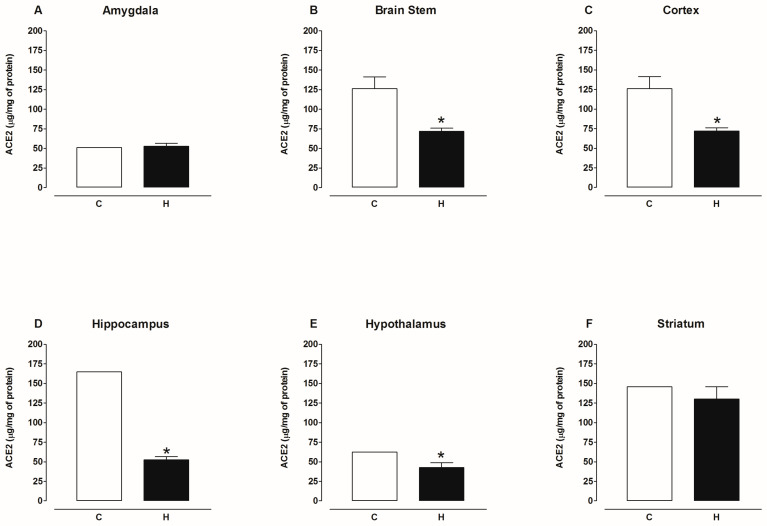
ACE2 content in amygdala, brain stem, cortex, hippocampus, hypothalamus, and striatum of control (C) and hypertensive (H). Each bar represents the $\bar{x}$ ± SE of *n* = 10. * *p* < 0.05 when compared with control.

## Data Availability

Data is contained within the article and Appendix A.

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
