# Peer review of "SARS-CoV-2 Spike Protein Enhances Carboxypeptidase Activity of Angiotensin-Converting Enzyme 2"

_ijms, 2024, doi:10.3390/ijms25116276_

Round 1
Reviewer 1 Report
Comments and Suggestions for Authors
The authors analyzed the ACE2 activity in plasma and lysates of heart, kidney, liver, lung, and six brain regions in diabetic and hypertensive rats and how their activity ca be affected by the SARS-CoV-2 spike protein. The authors demonstrated that ACE2 protein increased in the heart, while decreased in the plasma, kidney, hypothalamus, hippocampus, striatum, cortex, and brain stem of diabetic rats. The ACE2 protein decreased in the hypothalamus, hippocampus, cortex, and brain stem of hypertensive rats. The topic is interesting and important, please see the suggestions below.
1. Please add several sentences to show the importance of the study, in the introduction section.
2. The authors measured ACE2 activity with and without 10 μg of spike protein. More information is needed. Is the whole protein used in the assay? How was the protein expressed? The spike protein of What strain? Why use 10ug of spike protein?
3. Line 266, if the spike protein enhances the ACE2 activity, then the metabolites generated by ACE2 are good or bad. This sentence is confusing.
Author Response
Thank you very much for your comments.

Reviewer 2 Report
Comments and Suggestions for Authors
Mendiola-Salazar et al. suggest that SARS-CoV-2 spike protein can affect the activity of ACE2 in sevral organs of diabetic and hypertensive rats. They indirectly measured the ACE2 activity by a fluorescence-based assay. Although they confirmed that this assay works with ACE2 inhibitors, the impact of spike protein on rat ACE2 activity does not seem to be large.
1. Since the differences between with and without spike protein are small, I think the authors should show the same effect of spike protein with different experiments too. At least, they need to show dose dependency of spike protein on the ACE2 activity. They used 10 ug spike protein which is very large amount, so they can use several smaller amounts for showing dose dependency.
2. As rats are not susceptible for SARS-CoV-2 infection, I am not able to find any interest in this regulation of rat ACE2 activity by the spike protein. The authors also cannot confirm their conclusion with SARS-CoV-2 infection instead of treatment of 10 ug spike protein.
Author Response
Thank you very much for your comments.

Round 2
Reviewer 1 Report
Comments and Suggestions for Authors
this is an updated version and the authors have taken all comments and suggestions into consideration, the manuscript is improved. I did not detect any caveat raised by the present study to be addressed.
Author Response
Thank you very much for your comments.

Reviewer 2 Report
Comments and Suggestions for Authors
Mendiola-Salazar et al. suggest that SARS-CoV-2 spike protein can affect the activity of ACE2 in sevral organs of diabetic and hypertensive rats. They indirectly measured the ACE2 activity by a fluorescence-based assay. Although they confirmed that this assay works with ACE2 inhibitors, the impact of spike protein on rat ACE2 activity does not seem to be large.
1. Since the differences between with and without spike protein are small, I think the authors should show the same effect of spike protein with different experiments too. At least, they need to show dose dependency of spike protein on the ACE2 activity. They used 10 ug spike protein which is very large amount, so they can use several smaller amounts for showing dose dependency.
2. As rats are not susceptible for SARS-CoV-2 infection, I am not able to find any interest in this regulation of rat ACE2 activity by the spike protein. The authors also cannot confirm their conclusion with SARS-CoV-2 infection instead of treatment of 10 ug spike protein.
Author Response
I apologize for not uploading the file with the response to your comments.
We are sending a new version of the paper considering your comments.

Round 3
Reviewer 2 Report
Comments and Suggestions for Authors
They have addressed all my concerns, performing an additional experiment and mentioning a limitation in this study.